# CoSyn: Detecting Implicit Hate Speech in Online Conversations Using a Context Synergized Hyperbolic Network

**Sreyan Ghosh**[1*]    **Manan Suri**[2,3*]    **Purva Chiniya**[1*]    **Utkarsh Tyagi**[1*]
**Sonal Kumar**[1*]    **Dinesh Manocha**[1]
[1]University of Maryland College Park, USA, [2]MIDAS Labs, IIIT-Delhi, India
{sreyang, pchiniya, utkarsht, sonalkum, dmanocha}@umd.edu,
manansuri27@gmail.com

## Abstract

The tremendous growth of social media users interacting in online conversations has led to significant growth in hate speech affecting people from various demographics. Most of the prior works focus on detecting *explicit hate speech*, which is overt and leverages hateful phrases, with very little work focusing on detecting hate speech that is *implicit* or denotes hatred through indirect or coded language. In this paper, we present **CoSyn**, a **co**ntext **syn**ergized neural network that explicitly incorporates user- and conversational-context for detecting *implicit hate speech* in online conversations. CoSyn introduces novel ways to encode these external contexts and employs a novel context interaction mechanism that clearly captures the interplay between them, making independent assessments of the amounts of information to be retrieved from these noisy contexts. Additionally, it carries out all these operations in the hyperbolic space to account for the scale-free dynamics of social media. We demonstrate the effectiveness of CoSyn on 6 hate speech datasets and show that CoSyn outperforms all our baselines in detecting implicit hate speech with absolute improvements in the range of 1.24% - 57.8%. We make our code available[1].

## 1 Introduction

Hate speech is defined as the act of making utterances that can potentially offend, insult, or threaten a person or a community based on their caste, religion, sexual orientation, or gender (Schmidt and Wiegand, 2019). For social media companies like Twitter, Facebook, and Reddit, appropriately dealing with hate speech has been a crucial challenge (Tiku and Newton, 2015). Hate speech can take the form of overt abuse, also known as *explicit hate speech* (Schmidt and Wiegand, 2017), or can be uttered in coded or indirect language, also known as

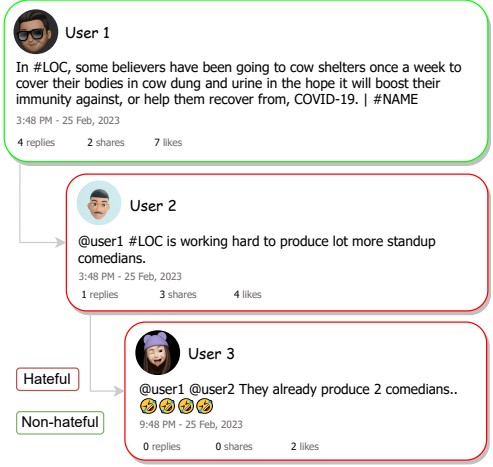

Figure 1: Illustration of a social media conversation tree with *implicit hate speech*. User 1 posts a factual statement about practices people follow in a certain location. In response, User 2 implies hate through a sarcastic statement, to which User 3 elaborates with a positive stance. Clearly, these utterances are even difficult for humans to classify as hate or not without the proper conversational context.

*implicit hate speech* (Jurgens et al., 2019). Fig.1 illustrates a Twitter conversation where users convey hate implicitly through sarcasm and conversational context.

**Societal Problem and Impact.** Though a considerable amount of research has been done on detecting explicit hate speech (Schmidt and Wiegand, 2019), detecting implicit hate speech is a greatly understudied problem in the literature, despite the social importance and relevance of the task. In the past, extremist groups have used coded language to assemble groups for acts of aggression (Gubler and Kalmoe, 2015) and domestic terrorism (Piazza, 2020) while maintaining deniability for their actions (Dénigot and Burnett, 2020). Because it lacks clear lexical signals, implicit hate utterances evade keyword-based detection systems (Wiegand et al., 2019), and even the most advanced neural architectures may not be effective in detecting such utter-

---

[1]https://github.com/Sreyan88/CoSyn
*These authors contributed equally to this work.

ances (Caselli et al., 2020).

**Current Challenges.** Current state-of-the-art hate speech detection systems fail to effectively detect implicit and subtle hate (Ocampo et al., 2023). Detecting implicit hate speech is difficult for multiple reasons: **(1)** Linguistic nuance and diversity: Implicit hate can be conveyed through sarcasm, humor (Waseem and Hovy, 2016), euphemisms (Magu and Luo, 2018), circumlocution (Gao and Huang, 2017), and other symbolic or metaphorical languages (Qian et al., 2018). **(2)** Varying context: Implicit hate can be conveyed through everything from dehumanizing comparisons (Leader Maynard and Benesch, 2016), and stereotypes (Warner and Hirschberg, 2012) to threats, intimidation, and incitement to violence (Sanguinetti et al., 2018; Fortuna and Nunes, 2018). **(3)** Lack of sufficient linguistic signals: Unlike parent posts, which contain sufficient linguistic cues through background knowledge provided by the user, replies or comments to the parent post are mostly short and context-less reactions to the parent post. These factors make implicit hate speech difficult to detect and emphasize the need for better learning systems.

**Why prior work is insufficient.** (ElSherief et al., 2021) define implicit hate speech as "coded or indirect language that disparages a person or group." They also propose the first dataset, *Latent Hatred*, to benchmark model performance on implicit hate speech classification and show that existing state-of-the-art classifiers fail to perform well on the benchmark. Though *Latent Hatred* builds on an exhaustive 6-class taxonomy, it ignores implicit hate speech that is conversational-context-sensitive even though it accounts for a majority of implicit hate speech online (Modha et al., 2022; Hebert et al., 2022). Lin (2022) builds on Latent Hatred and propose one of the first systems to classify implicit hate speech leveraging world knowledge through knowledge graphs (KGs). However, beyond the fact that their system is restricted to only English due to the unavailability of such KGs in other languages, their system also fails to capture any kind of external context, which is vital for effective hate speech detection (Sheth et al., 2022). Thus, we first extend the definition of implicit hate speech to include utterances that convey hate only in the context of the conversational dialogue (example in Fig.1). Next, we propose a novel neural learning system to solve this problem.

**Main Contributions.** In this paper, we propose CoSyn, a novel neural network architecture for detecting implicit hate speech that effectively incorporates external contexts like conversational and user. Our primary aim is to classify whether a target utterance (text only) implies hate or not, including the ones that signal hate implicitly. CoSyn jointly models the user's personal context (historical and social) and the conversational dialogue context in conversation trees. CoSyn has four main components: **(1)** To encode text utterances, we train a transformer sentence encoder and promote it to learn bias-invariant representations. This helps us to handle keyword bias, a long-standing problem in hate speech classification (Garg et al., 2022). **(2)** We start by modeling the user's *personal historical context* using a novel **Hyperbolic Fourier Attention Network (HFAN)**. HFAN models diverse and scale-free user engagement of a user on social media by leveraging Discrete Fourier Transform (Cooley and Tukey, 1965) and hyperbolic attention on past user utterances. **(3)** We next model the user's *personal social context* using a **Hyperbolic Graph Convolutional Network (HGCN)** (Chami et al., 2019). HGCN models the scale-free dynamics of social networks using hyperbolic learning, leveraging the social connections between users, which act as edges in the graph. **(4)** Finally, to *jointly model a user's personal context and the conversational dialogue context*, we propose a novel **Context Synergized Hyperbolic Tree-LSTM (CSHT)**. CSHT effectively models the scale-free nature of conversation trees and clearly captures the interaction between these context representations in the hyperbolic learning framework. We describe all our components in detail in Section 2.7. To summarize, our main contributions are as follows:

- We introduce CoSyn, the first neural network architecture specifically built to detect implicit hate speech in online conversations. CoSyn leverages the strengths of existing research and introduces novel modules to explicitly take into account user and conversational context integral to detecting implicit hate speech.

- Through extensive experimentation, we show that CoSyn outperforms all our baselines quantitatively on 6 hate speech datasets with absolute improvements of 1.24% - 57.8%.

- We also perform extensive ablative experiments and qualitative comparisons to prove

the efficacy of CoSyn.

## 2 Methodology

Fig. 3 provides a clear pictorial representation of our proposed model architecture (algorithm shown in Algorithm 1). We provide an overview describing various operations in Hyperbolic Geometry in Appendix 2.2; and we request our readers to refer to that for more details. In this section, we describe the three constituent components of our proposed **CoSyn**, which model different aspects of context, namely, the author's personal historical context, the author's personal social context, and both the personal historical and social contexts with the conversational context.

### 2.1 Background, Notations, and Problem Formulation.

Let's suppose we have $N$ distinct conversation trees, where each tree is denoted by $T_n \in T = \{T_0, \cdots, T_n, \cdots, T_N\}$. Each tree $T_n$ has $J^{T_n}$ individual utterances denoted by $T_n = \{t_0^{T_n}, \cdots, t_j^{T_n}, \cdots, t_J^{T_i}\}$. Each individual utterance $t_j^{T_n}$ is authored by one of the $L$ users in the dataset denoted by $u \in \{u_0, \cdots, u_l, \cdots, u_L\}$. The primary aim of CoSyn is to classify each utterance $t$ into its label $y^{gt} \in \{0, 1\}$ where 0 and 1 indicate whether the utterance is hateful or not. Additionally, each hateful utterance ($y_i^{gt} = 1$) is labeled with $y^{ip} \in \{0, 1\}$ where 0 and 1 indicate whether the hateful utterance is explicitly or implicitly hateful. $y^{ip}$ is used only for analysis. In the following subsections, "user" refers to the author of an utterance in a conversation tree to be assessed.

### 2.2 Hyperbolic Geometry: Background

**Hyperbolic Geometry.** A Riemannian manifold is a $D$-dimensional real and smooth manifold defined with an inner product on tangent space $g_x : \mathcal{T}_x \mathcal{M} \times \mathcal{T}_x \mathcal{M} \to \mathbb{R}$ at each point $x \in \mathcal{M}$, where the tangent space $\mathcal{T}_x \mathcal{M}$ is a $D$-dimensional vector space representing the first-order local approximation of the manifold around a given point $x$. A Poincaré ball manifold is a hyperbolic space defined as a constant negative curvature Riemannian manifold, denoted by $(\mathbb{H}^D, g)$ where manifold $\mathbb{H}^D = \{x \in \mathbb{R}^D : \|x\| < 1\}$ and the Riemannian metric is given by $g_x = \lambda_x^2 g^E$ where $g^E = I_D$ denotes the identity matrix, i.e., the Euclidean metric tensor and $\lambda_x = \frac{1}{1 - \|\|x\|\|^2}$. To perform operations in the hyperbolic space, the exponential map

$\exp_x : \mathcal{T}_x \mathbb{H}^D \to \mathbb{H}^D$ and the logarithmic map $\log_x : \mathbb{H}^D \to \mathcal{T}_x \mathbb{H}^D$ are used to project Euclidean vectors to the hyperbolic space and vice versa respectively.

$$\exp_x(v) := x \oplus \left( \tanh\left( \frac{\lambda_x \|v\|}{2} \right) \frac{v}{\|v\|} \right) \quad (1)$$

$$\log_x(y) := \frac{2}{\lambda_x} \tanh^{-1}(\|-x \oplus y\|) \frac{-x \oplus y}{\|-x \oplus y\|} \quad (2)$$

where $x, y \in \mathbb{H}^D$, $v \in \mathcal{T}_x \mathbb{H}^D$. Further, to perform operations in the Hyperbolic space, the following basic hyperbolic operations are described below:

**Möbius Addition** $\oplus$ adds a pair of points $x, y \in \mathbb{H}^D$ as,

$$x \oplus y := \frac{(1 + 2\langle x, y \rangle + \|y\|^2) x + (1 - \|x\|^2) y}{1 + \langle x, y \rangle + \|x\|^2 \|y\|^2} \quad (3)$$

**Möbius Multiplication** $\otimes$ multiplies vectors $x \in \mathbb{H}^D$ and $W \in \mathbb{R}^{D' \times D}$ given by,

$$\mathbf{W} \otimes x := \exp_o (\mathbf{W} \log_0(x)) \quad (4)$$

**Möbius Element-wise Multiplication** $\odot$ performs element-wise multiplication on $x, y \in \mathbb{H}^D$,

$$x \odot y := \tanh\left( \frac{\|xy\|}{y} \tanh^{-1}(\|y\|) \right) \frac{\|xy\|}{\|y\|} \quad (5)$$

**Hyperbolic Distance** between points $x, y \in \mathbb{H}^D$ is given by:

$$d_\mathcal{B} := 2 \tanh^{-1}(\|-x \oplus y\|) \quad (6)$$

**Fourier Transform.** The Fourier transform breaks down a signal into its individual frequency components. When applied to a sequence $x_n$ with $n \in [0, N-1]$, the Discrete Fourier Transform (DFT) generates a new representation $X_k$ for each value of k, where $0 \leq k \leq N-1$. DFT accomplishes this by computing a sum of the original input tokens $x_n$, multiplied by twiddle factors $[e^{-2\pi i k n / N}]$, where $n$ is the index of each token in the sequence. Thus, DFT is expressed as:

$$X_k = \sum_{n=0}^{N-1} x_n e^{-2\pi i k n/N}, \quad 0 \le k \le N-1 \quad (7)$$

## 2.3 Bias-invariant Encoder Training

CoSyn, like other works in literature, builds on the assumption that linguistic signals from text utterances can effectively enable the detection of hate speech in online conversations. Thus, our primary aim is to learn an encoder $e(.)$ that can effectively generate vector representations $\mathbb{R}^d$ for a text utterance where $d$ is the dimension of the vector. Specifically, we fine-tune a SentenceBERT (Reimers and Gurevych, 2019) and solve an additional loss proposed in (Mathew et al., 2021) using self-attention maps and ground-truth hate spans. Keyword bias is a long-standing problem in hate speech classification, and solving the extra objective promotes learning bias-invariant sentence representations.

## 2.4 Modeling Personal User Context

Modeling personal user context, in the form of author profiling, for hate speech classification has shown great success in the past because hateful users (users prone to making hate utterances) share common stereotypes and form communities around them. They exhibit strong degrees of *homophily* and have high reciprocity values (Mathew et al., 2019). We hypothesize that this will prove to be especially useful in classifying *conversational implicit hate speech*, which on its own lacks clear lexical signals or any form of background knowledge. Thus, our primary aim is to learn a vector representation $\mathcal{U}_u \in \mathbb{R}^d$ for the user $u$ who has authored the utterance $t$ to be assessed, where $d$ is the dimension of the vector. For our work, we also explore the importance of the personal historical context of a user to enable better author profiling. Studies show that past social engagement and linguistic styles of their utterances on social media platforms play an important role in assessing the user's ideology and emotions (Xiao et al., 2020). Thus we propose a novel methodology for author profiling that is more intuitive, explainable, and effective for our task. Our primary aim is to model the user's (author of the utterance to be assessed) personal context and generate user representations $\mathcal{U}_u$, which can then be used for hate speech classification. To achieve this, we first encode a user's historical utterances $\mathcal{U}_u^{hist}$ using our Hyperbolic

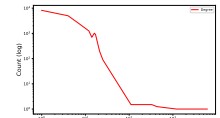
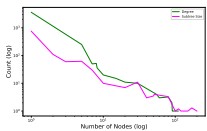

(a) Frequency distribution for the social graph.

(b) Frequency distribution for conversation trees.

| Property | Social Graph | Conversation Tree |
|---|---|---|
| Mean Degree | 18.22 | 2.2 |
| Node Degree | 2.66E-03 | 2.89E-04 |
| $\delta$ | 1.5 | 0.3 |
| Power Law | $P(x) \sim x^{-\gamma}$ | |
| $\gamma$ | 2.81 | 2.39 |

(c) Properties of the social graph and conversation trees.

Figure 2: Various properties of the social graph and conversation trees averaged across datasets. The fitting of the node distribution in the power law ($\gamma \in [2,3]$) (Choromański et al., 2013) and low hyperbolicity (Barabási and Bonabeau, 2003) $\delta$ indicates the scale-free nature of conversations and social graphs.

Fourier Attention Network (HFAN) followed by modeling the user's social context by passing $\mathcal{U}_u^{hist}$ through a Hyperbolic Graph Convolution Network (HGCN). To be precise, learning personal user context takes the form of $e(HistoricalUtterances)$ $\rightarrow$ HFAN $\rightarrow \mathcal{U}_u^{hist} \rightarrow$ HGCN $\rightarrow \mathcal{U}_u$. We next describe, in detail, HFAN and HGCN.

## 2.5 Hyperbolic Fourier Attention Network

User engagement on social media is often diverse and possesses scale-free properties (Sigerson and Cheng, 2018). Thus to account for the natural irregularities and effectively model a user's personal historical context, we propose a novel Hyperbolic Fourier Attention Network (**HFAN**). The first step is to encode historical utterances, $H^u$, using our encoder $e(.)$, made by the user $u$, denoted by $H^u \in \{h_0^u, \cdots, h_s^u, \cdots, h_S^u\}$. Next, we apply 2D DFT to the embeddings (1D DFT along the temporal dimension and 1D DFT along the embedding dimension) to capture the most commonly occurring frequencies (ideologies, opinions, emotions, etc.) in the historical utterances made by the user using:

$$\mathcal{U}_u^{fourier} = \exp_{\mathbf{0}}^c \left( \mathcal{F}_s \left( \mathcal{F}_h \left( \log_0^c \left( e(H^u) \right) \right) \right) \right) \quad (8)$$

where $\mathcal{F}$ is the DFT operation. The 2D DFT operation helps highlight the most prominent frequencies, which signifies a holistic understanding of the user's sociological behavior. Next, we hypothesize that the frequencies most relevant to a conversation can act as the most important clues to

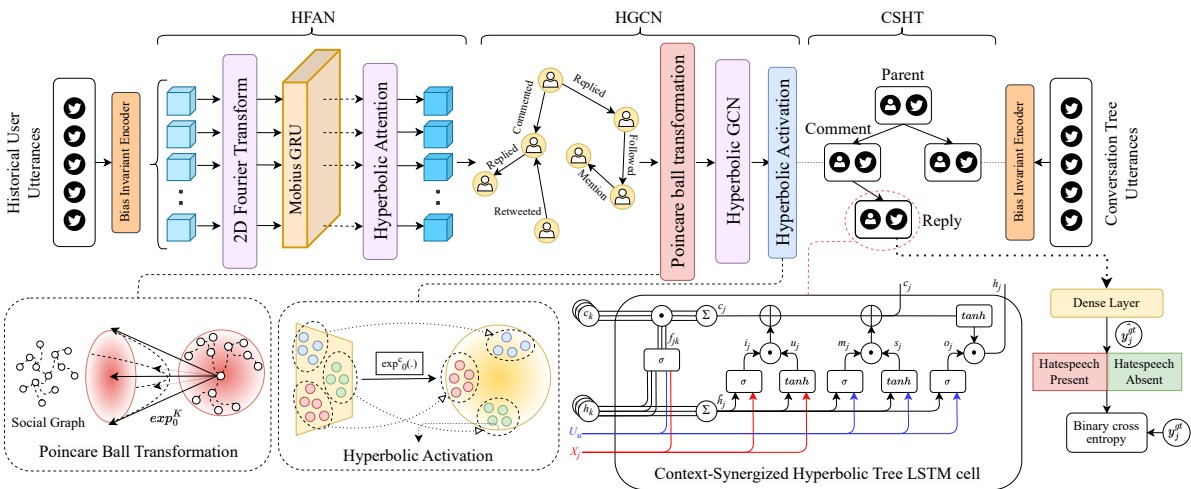

Figure 3: Illustration of **CoSyn**. CoSyn detects if a target utterance, authored by a social media user, implies hate or not leveraging three main components: (1) **HFAN**: This models the user's personal historical context by employing Fourier Transform and Hyperbolic Attention on the user's historical tweets. (2) **HGCN**: This models the user's social context through hyperbolic graph learning on its relations with other users. (3) **CSHT**: This jointly models the user's personal context and the conversational context to finally classify if the utterance is hateful.

assessing how a user would react to other utterances in the conversation tree (for e.g., a user's stance on the contribution of a particular political party to a recent riot in discussion). Additionally, these frequencies may change over time. Thus, to account for the latter factor first, we pass the embeddings through a Hyperbolic GRU (Zhang and Gao, 2021), which first projects these embeddings from the Euclidean to the hyperbolic space using a Poincaré ball manifold and then effectively captures the sequential temporal context on time-aligned historic user utterances. The Hyperbolic GRU is a modified version of the conventional GRU, which performs Mobius operations on the Poincaré model (addition, multiplication, and bias). We request our readers to refer to (Zhang and Gao, 2021) for more details. Next, to account for the former factor, we perform Hyperbolic Attention (Zhang and Gao, 2021), which first linearly projects the utterance embeddings within Poincaré space and then constructs the final user embedding $\mathcal{U}_u^{hist}$ via Einstein midpoint:

$$\alpha_i = \exp\left(-\beta_h d_{\mathcal{L}}\left(\mathbf{c}_{h\mathcal{L}}, {\mathbf{h}_{s\mathcal{L}}^u}'\right) - c_h\right) \quad (9)$$

$$\mathcal{U}_u^{hist} = \sum_S \left[\frac{\alpha_i \gamma\left(\mathbf{h}_{i\mathcal{K}}^u\right)}{\sum_l \alpha_l \gamma\left(\mathbf{h}_{i\mathcal{K}}^u\right)}\right] \mathbf{h}_{i\mathcal{K}}^u, \quad (10)$$

where $\mathbf{h}_{s\mathcal{L}}$ and $\mathbf{h}_{s\mathcal{K}}$ are the utterance encoding of utterance $h_s$ obtained from $\mathcal{U}_u^{fourier}$ projected from the Poincaré space into the Lorentz space and Klein space for computational convenience (Zhang

and Gao, 2021). $\mathbf{c}_{s\mathcal{L}}$ is the sentence centroid, which is randomly initialized and learnable.

## 2.6 Hyperbolic Graph Convolutional Network

After obtaining the user representations $\mathcal{U}_u^{hist}$ for every user $u$ in the dataset, we model social relationships between the users as a graph $\mathcal{G} = (\mathcal{V}, \mathcal{E})$. Each edge $e = \{u_i, u_j\} \in \mathcal{E}$ represents one of the four types of relations: 1) User $u_i$ retweets a tweet posted by $u_j$, 2) User $u_i$ mentions user $u_j$ in a tweet $t_t$, 3) User $u_i$ replies to a post by user $u_j$ or 4) User $u_i$ follows user $u_j$. Each vertex $v \in \mathcal{V}$ represents the user representations $\mathcal{U}_u$. **HGCN** (Chami et al., 2019) modifies the conventional GCN and performs neighbor aggregation using graph convolutions in the hyperbolic space to enrich a user's historical context representations learned through HFAN using social context. Social network connections between users on a platform often possess hierarchical and scale-free structural properties (degree distribution of nodes follows the power law as seen in Fig 2 and decreases exponentially with a few nodes having a large number of connections (Barabási and Bonabeau, 2003)). To model such complex hierarchical representations, HGCN performs all operations in the Poincaré space. We first project our representations $\mathcal{U}$ onto the hyperbolic space using a Poincaré ball manifold ($exp^K(.)$) with a sectional curvature -1/K. Formally, the feature aggregation based at the $i^{th}$ HGCN layer is denoted by:

$$\mathbf{O}^{(i)} = \sigma^{\otimes^{K_{i-1},K_i}}\left(\text{FM}\left(\tilde{\mathbf{A}}\mathbf{O}^{(i-1)}\mathbf{W}^{(i)}\right)\right) \quad (11)$$

where $-1/K_{i-1}$ and $-1/K_i$ are the hyperbolic curvatures at layers $i$ - 1 and $i$, respectively. $\tilde{\mathbf{A}} = D^{-1/2}AD^{-1/2}$ is the degree normalized adjacency matrix, $W$ is a trainable network parameter, $\mathbf{O}^i$ is the output of the $i^{th}$ layer and FM is the Frechet Mean operation. $\sigma^{\otimes^{K_{i-1},K_i}}$ is the hyperbolic non-linear activation allowing varying curvature at each layer.

## 2.7 Context-Synergized Hyperbolic Tree-LSTM

To model the conversational context in conversation trees effectively, we propose Context-Synergized Hyperbolic Tree-LSTM (**CSHT**). CSHT presents several modifications and improvements over Tree-LSTM (Tai et al., 2015), including (1) incorporating both the user's personal context and the conversation context while clearly capturing the interactions between them, and (2) operating in the hyperbolic space, unlike the original TreeLSTM, which operates in the Euclidean space. Conversation trees on social media possess a hierarchical structure of message propagation, where certain nodes may have many replies; e.g., nodes that include utterances from popular users. Such phenomena lead to the formation of hubs within the conversation tree, which indicates scale-free and asymmetrical properties of the conversation tree (Avin et al., 2018). The conversation tree is $T$ represented using $\mathcal{T} = (\mathcal{V}, \mathcal{E})$, where vertex $v \in \mathcal{V}$ represents the encoded utterance $\mathcal{X}_j = \text{e}(t_j)$ ($t$ is part of the conversation tree $T$) and edge $e \in \mathcal{E}$ represents one of the three relations between the utterances: (1) parent-comment, (2) comment-reply or (3) reply-reply. CSHT is modeled as a Bi-Directional Tree-LSTM $\left[\overrightarrow{\text{CSHT}}\left(\mathcal{X}_j;\mathcal{U}_u\right), \overleftarrow{\text{CSHT}}\left(\mathcal{X}_j;\mathcal{U}_u\right)\right]$ where $\mathcal{U}_u$ is the user representation of the user $u$ who authored the utterance to be assessed obtained from the HGCN.

In order to understand the signals in CHST, we focus on a specific node $t_j$. We gather information from all input nodes $t_k$ where $\{t_k, t_j\} \in \mathcal{E}$, and combine their hidden states $h_k$ using Einstein's midpoint to produce an aggregated hidden state $\widetilde{h_j}$ for node $t_j$. We use the hyperbolic representation of the current post, denoted as $\mathcal{X}_j$, as well as the user embeddings of the author of the post, denoted as $\mathcal{U}_u$, to define computational gates operating in the hyperbolic space for CSHT cells. As defined in Subsection 2.2 $\odot, \oplus, \otimes$ represent Möbius Dot Product, Möbius Addition and Möbius matrix multiplication, respectively.

Considering multiple input nodes $t_k, \{t_k, t_j\} \in \mathcal{E}$, we implement multiple hyperbolic forget gates incorporating the outputs $h_k$ of the input nodes with the current node's combined user and post representation $r_j$.

$$r_j = \mathbf{W}^{(\mathbf{fx})} \otimes \mathcal{X}_j \oplus \mathbf{W}^{(\mathbf{fg})} \otimes \mathcal{U}_u,$$
$$f_{jk} = \exp_o\left(\sigma\left(\log_o\left(r_j \oplus \mathbf{U}^{(\mathbf{f})} \otimes h_k \oplus \mathbf{b}^{(\mathbf{f})}\right)\right)\right) \quad (12)$$

The input gate $i_j$ and the intermediate memory gate $u_j$ corresponding to the representation for the current utterance are given by:

$$i_j = \exp_o\sigma\left(\log_o\left(\mathbf{W}^{(\mathbf{i})} \otimes \mathcal{X}_j \oplus \mathbf{U}^{(\mathbf{i})} \otimes \right.\right.$$
$$\left.\left. \tilde{h}_j \oplus \mathbf{b}^{(\mathbf{i})}\right)\right) \quad (13)$$
$$u_j = \exp_o\left(\tanh\left(\log_o\left(\mathbf{W}^{(\mathbf{u})} \otimes \mathcal{X}_j \oplus \mathbf{U}^{(\mathbf{u})} \otimes \right.\right.\right.$$
$$\left.\left.\left. \tilde{h}_j \oplus \mathbf{b}^{(\mathbf{u})}\right)\right)\right) \quad (14)$$

The input gate $m_j$ and the intermediate memory gate $s_j$ corresponding to the user representation from the social graph are given by:

$$m_j = \exp_o\left(\sigma\left(\log_o\left(\mathbf{W}^{(\mathbf{m})} \otimes \mathcal{U}_u \oplus \right.\right.\right.$$
$$\left.\left.\left. \mathbf{U}^{(\mathbf{m})} \otimes \widetilde{h}_j \oplus \mathbf{b}^{(\mathbf{m})}\right)\right)\right) \quad (15)$$
$$s_j = \exp_o\left(\tanh\left(\log_o\left(\mathbf{W}^{(\mathbf{s})} \otimes \mathcal{U}_u \oplus \right.\right.\right.$$
$$\left.\left.\left. \mathbf{U}^{(\mathbf{s})} \otimes \widetilde{h}_j \oplus \mathbf{b}^{(\mathbf{s})}\right)\right)\right) \quad (16)$$

The output gate for the tree cell is then calculated as follows:

$$o_j = \exp_o\left(\sigma\left(\log_o\left(\mathbf{W}^{(\mathbf{o})} \otimes \mathcal{U}_u \oplus \mathbf{U}^{(\mathbf{o})} \otimes \widetilde{h}_j \oplus \mathbf{b}^{(\mathbf{o})}\right)\right)\right) \quad (17)$$

The parameters $\mathbf{W}^{(\mathbf{w})}, \mathbf{U}^{(\mathbf{w})}, \mathbf{b}^{(\mathbf{w})}$ are learnable parameters in the respective gate $\mathbf{w}$. We obtain the cell state for the current cell $c_j$ by combining the representations from the multiple forget gates $f_{jk}$, $k \forall t_k, s.t.\{t_k, t_j\} \in \mathcal{E}$, as well as gates corresponding to the tweet and user representations as follows:

$$c_j = i_j \odot u_j \oplus m_j \odot s_j \oplus \sum_k f_{jk} \odot c_k \quad (18)$$

These equations are applied recursively by blending in tree representations at every level. The output of the current cell, $h_j$ is given by:

$$\text{CHST}(\mathcal{X}_j, \mathcal{U}_j) = h_j = o_j \odot \exp_o (\tanh (c_j))$$
$$(19)$$

**Final Prediction Layer.** We concatenate the node output from CSHT $h_j$, projected to the Euclidean space, with the Euclidean utterance features for the given tweet, $\mathcal{X}_j$ to form a robust representation incorporating features of the tweet along with social and conversational context derived from the CSHT network of CoSyn. This concatenated representation is passed through a final classification layer to obtain the final prediction $y_j^{gt} = \text{Softmax} (\text{MLP} ([\log_o (h_j) ; \mathcal{X}_j]))$. For training, we optimize a binary cross-entropy loss.

## 3 Experiments and Results

### 3.1 Dataset

We evaluate CoSyn on 6 conversational hate speech datasets; namely, Reddit (Qian et al., 2019), GAB (Qian et al., 2019), DIALOCONAN (Bonaldi et al., 2022), CAD (Vidgen et al., 2021), ICHCL (Modha et al., 2021) and Latent Hatred (ElSherief et al., 2021). Appendix A provides dataset statistics. Table 1 reports the micro-$F_1$ scores averaged across 3 runs with 3 different random seeds.

**Implicit Hate Annotations.** The original datasets do not have annotations to indicate if an utterance denotes implicit or explicit hate speech. Thus, for our work, we add extra annotations for evaluating how CoSyn fares against our baselines in understanding the context for detecting implicit hate speech. Specifically, we use Amazon Mechanical Turk (MTurk) and ask three individual MTurk workers to annotate if a given utterance conveys hate implicitly or explicitly. The primary factor was the requirement of conversational context to understand the conveyed hate. At stage (1), we provide them with the definition of hate speech and examples of explicit and implicit hate speech. At stage (2), we provide complete conversations and ask them to annotate implicit or explicit in a binary fashion. Cohen's Kappa Scores for inter-annotator agreement are provided with the dataset details in Appendix A.

### 3.2 Baselines

Due to the acute lack of prior work in this space, beyond just hate speech classification, we also compare our method with systems proposed in the thriving fake news detection literature, which considers conversational and user contexts to detect fake news. Some of these systems had to be adapted

to our task, and we describe about the working of each baseline in detail in Appendix B. Specifically, we compare CoSyn with Sentence-BERT (Reimers and Gurevych, 2019), ConceptNet, HASOC (Farooqi et al., 2021), Conc-Perspective (Pavlopoulos et al., 2020), CSI (Ruchansky et al., 2017), GCAN (Lu and Li, 2020), HYPHEN (Grover et al., 2022), FinerFact (Jin et al., 2022), GraphNLI (Agarwal et al., 2022), DUCK (Tian et al., 2022), MRIL (Sawhney et al., 2022a) and Madhu (Madhu et al., 2023).

### 3.3 Hyperparameters

We decide on the best hyper-parameters for CoSyn based on the best validation set performance using grid search. The optimal hyperparameters are found to be, batch-size $b = 32$, HGCN output embedding dimension $g = 512$, hidden dimension of CHST $h = 768$, latent dimension of HFAN $l = 100$, learning rate $lr = 1.3e^{-2}$, weight decay $\beta = 3.2e^{-4}$, and dropout rate $\delta = 0.41$.

### 3.4 Results and Ablations

Table 1 compares the performance of CoSyn with all our baselines on 6 hate speech datasets. As we see, Cosyn outperforms all our baselines both on the entire dataset and on the implicit subset, thus showing its effectiveness for implicit hate speech detection. Despite the ambiguous nature of implicit hate speech, CoSyn has high precision scores for implicit hate speech, thus implying that it mitigates the problem of false positives better than other baselines. One common observation is that our bias invariant SentenceBERT approach emerges as the most competitive baseline to CoSyn, thereby reinforcing that most prior work do not effectively leverage external context.

When evaluated on the entire dataset, CoSyn achieves absolute improvements in the range of 5.1% - 35.2% on Reddit, 4.0% - 45.0% on CAD, 4.8% - 22.4% on DIALOCONAN, 3.9% - 42.7% on GAB, 9.7% - 38.4% on ICHCL and 5.8% - 31.6% on Latent Hatred ober our baselines. When evaluated on the implicit subsets, CoSyn achieves absolute improvements in the range of 5.1% - 57.9% on Reddit, 8.6% - 31.3% on CAD, 18.2% - 40.7% on DIALOCONAN, 2.9% - 28.5% on GAB, 8.2% - 19.5% on ICHCL and 1.2% - 18.1% on Latent Hatred.

Table 2 ablates the performance of CoSyn, removing one component at a time to show the significance of each. All results have been averaged

Table 1 — top section (datasets: Reddit, CAD, DIALOCONAN):

| | Overall F₁ | P | R | Implicit F₁ | P | R | C.F₁ | R.F₁ | Overall F₁ | P | R | Implicit F₁ | P | R | C.F₁ | R.F₁ | Overall F₁ | P | R | Implicit F₁ | P | R | C.F₁ | R.F₁ |
|---|---|---|---|---|---|---|---|---|---|---|---|---|---|---|---|---|---|---|---|---|---|---|---|---|
| **Baseline** | **Reddit** | | | | | | | | **CAD** | | | | | | | | **DIALOCONAN** | | | | | | | |
| SentenceBert | 71.12 | 63.35 | 81.06 | 76.05 | 77.01 | 75.11 | 22.45 | 18.06 | 46.89 | 51.30 | 43.18 | 49.01 | 48.03 | 50.03 | 35.40 | – | 46.23 | 37.20 | 60.70 | 34.75 | 26.46 | 50.60 | 49.98 | 29.55 |
| ConceptNet | 58.25 | 58.72 | 58.73 | 26.43 | 32.57 | 22.23 | 33.34 | 22.17 | 56.68 | 54.22 | 59.37 | 39.42 | 42.50 | 36.75 | 28.72 | – | 39.45 | 34.22 | 46.56 | 24.53 | 25.90 | 23.29 | 35.15 | 15.72 |
| HASOC | 56.21 | 51.86 | 61.35 | 28.9 | 31.48 | 26.71 | 35.50 | 20.35 | 50.11 | 47.54 | 52.97 | 40.61 | 46.75 | 35.89 | 30.29 | – | 43.67 | 32.45 | 66.74 | 27.22 | 23.80 | 31.78 | 28.45 | 17.02 |
| Conc-Perspective | 56.21 | 51.86 | 61.35 | 25.75 | 29.65 | 22.75 | 27.25 | 24.50 | 40.11 | 42.50 | 37.97 | 33.34 | 35.60 | 31.34 | 28.89 | – | 35.33 | 32.16 | 39.19 | 22.33 | 20.65 | 24.30 | 31.20 | 18.92 |
| CSI | 52.77 | 50.90 | 54.80 | 33.40 | 30.25 | 38.28 | 34.00 | 20.50 | 65.80 | 63.18 | 68.64 | 43.14 | 44.25 | 42.00 | 30.35 | – | 42.23 | 45.01 | 39.77 | 26.25 | 28.90 | 24.05 | 30.60 | 17.55 |
| GCAN | 54.75 | 57.94 | 51.89 | 23.25 | 30.00 | 18.98 | 13.13 | 15.50 | 69.31 | 71.00 | 67.69 | 44.34 | 45.60 | 43.13 | 33.78 | – | 31.33 | 31.10 | 32.67 | 31.33 | 32.80 | 29.98 | 34.70 | 18.60 |
| HYPHEN | 53.44 | 55.12 | 54.38 | 25.00 | 27.10 | 23.20 | 27.40 | 20.00 | 42.65 | 40.11 | 45.50 | 32.80 | 30.22 | 35.86 | 29.98 | – | 34.67 | 38.80 | 31.34 | 16.40 | 20.00 | 13.89 | 19.19 | 12.30 |
| FinerFact | 63.25 | 62.14 | 64.40 | 27.11 | 30.25 | 24.56 | 40.55 | 25.70 | 65.36 | 64.00 | 66.77 | 26.25 | 30.12 | 23.26 | 18.22 | – | 30.70 | 28.89 | 32.70 | 15.60 | 18.20 | 13.60 | 21.15 | 18.11 |
| Graph NLI | 41.04 | 57.12 | 32.02 | 26.03 | 42.50 | 19.76 | 55.01 | 45.00 | 28.25 | 68.10 | 17.82 | 47.43 | 45.66 | 49.27 | 22.75 | – | 32.35 | 31.44 | 33.31 | 24.11 | 21.89 | 26.83 | 39.02 | 16.50 |
| DUCK | 60.10 | 61.89 | 58.40 | 34.23 | 26.26 | 49.15 | 40.55 | 32.30 | 30.66 | 32.50 | 29.02 | 28.11 | 24.25 | 33.43 | 22.15 | – | 28.60 | 28.20 | 29.01 | 20.45 | 18.23 | 23.28 | 26.89 | 14.40 |
| HCN | 56.01 | 55.64 | 56.38 | 0.03 | 31.18 | 28.96 | 26.90 | 21.67 | 34.63 | 37.18 | 32.41 | 27.65 | 22.30 | 36.37 | 23.87 | – | 39.65 | 34.67 | 46.30 | 25.31 | 23.89 | 26.91 | 22.19 | 15.90 |
| MRIL | 58.91 | 57.30 | 60.61 | 31.76 | 30.12 | 33.67 | 32.05 | | 34.41 | 35.27 | 33.59 | 40.65 | 42.90 | 38.62 | 21.35 | – | 40.82 | 39.45 | 42.28 | 25.09 | 22.31 | 28.66 | 20.77 | 16.31 |
| Madhu | 70.58 | 65.28 | 76.82 | 76.79 | 77.79 | 75.81 | 27.39 | 23.34 | 50.47 | 55.72 | 46.13 | 51.77 | 50.67 | 52.91 | 36.63 | – | 46.45 | 40.06 | 55.26 | 36.96 | 29.44 | 49.64 | 47.62 | 33.82 |
| **CoSyn (ours)** | **76.23** | **79.07** | 73.58 | **81.12** | **80.15** | **82.12** | **60.23** | **59.12** | **73.26** | 70.07 | **76.75** | **57.59** | **59.27** | **56.01** | **38.19** | – | **51.02** | **52.92** | 49.25 | **52.98** | **54.67** | 51.39 | 48.77 | **54.00** |

Table 1 — bottom section (datasets: GAB, ICHCL, Latent Hatred):

| | Overall F₁ | P | R | Implicit F₁ | P | R | C.F₁ | R.F₁ | Overall F₁ | P | R | Implicit F₁ | P | R | C.F₁ | R.F₁ | Overall F₁ | P | R | Implicit F₁ | P | R | C.F₁ | R.F₁ |
|---|---|---|---|---|---|---|---|---|---|---|---|---|---|---|---|---|---|---|---|---|---|---|---|---|
| **Baseline** | **GAB** | | | | | | | | **ICHCL** | | | | | | | | **Latent Hatred** | | | | | | | |
| SentenceBert | 50.31 | 42.67 | 61.28 | 40.03 | **49.78** | 33.47 | 24.06 | 13.10 | 79.86 | 81.03 | 78.82 | 37.32 | 36.33 | 38.11 | 36.69 | 37.11 | 58.82 | 45.45 | **83.33** | 38.46 | 31.25 | 50.00 | 27.05 | – |
| ConceptNet | 47.82 | 48.72 | 46.95 | 19.29 | 25.23 | 15.61 | 16.12 | 10.12 | 69.11 | 68.23 | 70.01 | 28.29 | 28.27 | 28.31 | 28.91 | 27.89 | 48.12 | 46.87 | 49.43 | 37.23 | 34.45 | 40.49 | 22.56 | – |
| HASOC | 39.45 | 43.44 | 36.13 | 16.54 | 22.11 | 13.21 | 15.19 | 12.71 | 72.53 | 72.67 | 72.51 | 34.31 | 35.09 | 33.56 | 35.28 | 32.11 | 50.47 | 52.22 | 48.83 | 39.82 | 37.22 | 42.81 | 35.21 | – |
| Conc-Perspective | 51.47 | 47.23 | 56.54 | 18.23 | 20.54 | 16.38 | 24.29 | 18.45 | 71.18 | 70.14 | 72.25 | 31.18 | 29.59 | 32.94 | 30.46 | 29.82 | 51.22 | 53.79 | 48.88 | 40.11 | 38.12 | 42.31 | 34.37 | – |
| CSI | 49.62 | 51.22 | 47.17 | 20.60 | 22.24 | 19.18 | 23.50 | 18.34 | 69.47 | 75.20 | 64.55 | 26.50 | 24.01 | 30.15 | 23.64 | 19.31 | 56.25 | 53.11 | 59.70 | 42.06 | 32.80 | 58.59 | 21.25 | – |
| GCAN | 24.00 | 32.00 | 27.00 | 26.47 | 30.25 | 23.53 | 17.20 | 15.36 | 74.22 | 72.11 | 76.45 | 35.22 | 36.20 | 34.29 | 32.14 | 26.47 | 56.80 | 54.55 | 59.24 | 40.24 | 31.65 | 55.23 | 20.71 | – |
| HYPHEN | 48.32 | 45.19 | 51.92 | 23.50 | 25.21 | 22.00 | 28.60 | 19.45 | 72.72 | 67.11 | 74.41 | 33.34 | 27.66 | 41.95 | 34.66 | 34.21 | 53.20 | 51.60 | 54.90 | 42.40 | **47.89** | 38.04 | 34.11 | – |
| FinerFact | 53.00 | 52.65 | 53.35 | 18.50 | 24.00 | 15.05 | 23.68 | 15.04 | 69.11 | 67.43 | 70.87 | 32.27 | 33.82 | 30.85 | 24.90 | 18.65 | 52.11 | 48.32 | 56.54 | 52.04 | 42.67 | **66.68** | 32.00 | – |
| Graph NLI | 50.00 | 53.22 | 47.15 | 42.12 | 45.00 | 39.58 | 47.00 | 32.00 | 51.17 | 42.98 | 63.22 | 26.53 | 21.29 | 35.19 | 27.65 | 28.15 | 33.10 | 25.24 | 48.07 | 48.25 | 39.24 | 62.63 | 33.00 | – |
| DUCK | 62.78 | 60.50 | 65.30 | 35.70 | 36.85 | 34.62 | 36.20 | 24.60 | 78.36 | 78.42 | 78.30 | 37.88 | 37.64 | 38.12 | 36.48 | 35.59 | 56.00 | 58.50 | 53.75 | 35.15 | 30.00 | 42.44 | 26.16 | – |
| HCN | 52.51 | 53.88 | 51.21 | 41.60 | 46.11 | 37.89 | 24.25 | 28.97 | 65.36 | 60.45 | 71.14 | 27.66 | 27.03 | 28.32 | 25.03 | 27.99 | 50.47 | 52.39 | 48.68 | 30.11 | 32.96 | 27.71 | 21.32 | - |
| MRIL | 61.21 | 60.75 | 61.67 | 40.92 | 33.65 | 52.19 | 42.01 | 33.89 | 66.71 | 67.87 | 65.59 | 30.22 | 31.45 | 28.91 | 29.01 | 29.89 | 57.32 | 54.88 | 59.98 | 40.16 | 37.80 | 42.83 | 37.61 | – |
| Madhu | 52.94 | 45.19 | 63.91 | 41.15 | 48.72 | 35.61 | 26.84 | 17.77 | 82.01 | 84.63 | 79.55 | 39.56 | 39.00 | 40.14 | 39.81 | 39.72 | 59.52 | 47.14 | 80.72 | 40.72 | 33.56 | 51.75 | 28.19 | – |
| **CoSyn (ours)** | **66.71** | **64.43** | **69.15** | 45.00 | 37.01 | 57.38 | 46.22 | **38.29** | **89.53** | **90.55** | **88.53** | **46.03** | **46.26** | **45.82** | **46.85** | **45.89** | **64.65** | **61.92** | 67.63 | **53.28** | 47.66 | 55.49 | **40.12** | – |

Table 1: Result comparison of CoSyn with our baselines on 6 hate speech datasets. We compare performance on both the overall dataset and the implicit subset. C.F₁ and R.F₁ indicate F₁ scores measured on only comments and replies, respectively. CoSyn outperforms all our baselines with absolute improvements in the range of 3.4% - 45.0% when evaluated on the entire dataset and 1.2% - 57.9% when evaluated on only the implicit subset. – indicates conversation trees in the dataset did not have replies.

across all 6 datasets. Some major observations include (1) CoSyn sees a steep drop in performance when used without user context (we model only the conversation context with a vanilla Tree-LSTM in the hyperbolic space). This proves the effectiveness of the user's personal context. Additionally, modeling user context with HFAN and HGCN proves to be more effective than just feeding mean-pooled historical utterance embeddings into CSHT. (2) Modeling in the Euclidean space results in an overall F1 drop of 3.8%. We reinforce the effectiveness of modeling in the hyperbolic space through our findings in Fig. 2.

| Ablations | Overall F₁ | P | R | Implicit F₁ | P | R | Comment F₁ | Reply F₁ |
|---|---|---|---|---|---|---|---|---|
| **CoSyn (ours)** | **70.23** | **69.83** | **70.82** | **56.00** | **54.17** | **58.04** | **46.73** | **49.32** |
| - DFT | 67.54 | 68.24 | 69.29 | 54.52 | 52.57 | 57.34 | 45.22 | 46.92 |
| - HFAN | 66.62 | 65.32 | 66.23 | 54.68 | 51.78 | 58.04 | 45.44 | 47.04 |
| - HGCN | 66.56 | 66.98 | 65.92 | 53.28 | 52.02 | 56.98 | 45.12 | 46.32 |
| - HFAN - HGCN | 65.29 | 64.71 | 65.55 | 52.14 | 51.12 | 56.38 | 42.91 | 46.29 |
| - User Context | 62.31 | 62.94 | 64.21 | 48.72 | 48.94 | 49.53 | 39.88 | 41.19 |
| BiCHST → UniCHST | 68.67 | 68.23 | 68.36 | 55.29 | 52.78 | 57.89 | 46.09 | 47.53 |
| Hyperbolic → Euclidean | 66.47 | 66.48 | 67.21 | 54.83 | 54.14 | 56.41 | 45.44 | 47.41 |

Table 2: Ablation study on Cosyn. Results are averaged across all 6 datasets.

## 3.5 Results Analysis

In Fig. 4, we show test set predictions from 4 different conversation trees from the ICHCL dataset with an attempt to study the effect of conversational and author context for target utterance classification. We notice that hateful users possess high degrees of homophily and are predominantly hateful over time. It also reveals a limitation of CoSyn where insufficient context from the parent leads to a false positive (comment in the 4th example).

## 4 Related Work

Prior research on identifying hate speech on social media has proposed systems that can flag online hate speech with remarkable accuracy (Schmidt and Wiegand, 2019). However, as mentioned earlier, prior efforts have focused primarily on classifying overt abuse or explicit hate speech (Schmidt and Wiegand, 2019) with little or no work on classifying implicit hate speech or that conveyed in coded language. The lack of research on this topic can also be attributed to existing datasets being skewed towards *explicitly* abusive text (Waseem and Hovy, 2016). Recently, this area of research has seen growing interest, with several datasets and benchmarks released for evaluating the performance of existing hate speech classifiers in identifying implicit hate speech (Caselli et al., 2020; ElSherief et al., 2021; Sap et al., 2019). One common observation is that most prior systems from literature fail to identify implicit hate utterances (ElSherief et al., 2021). Lin *et al.* (Lin, 2022) proposes one of the first systems to classify implicit hate speech leveraging world knowledge. However, they evaluate their performance on only *Latent Hatred*, which lacks conversational context. Additionally, acquiring

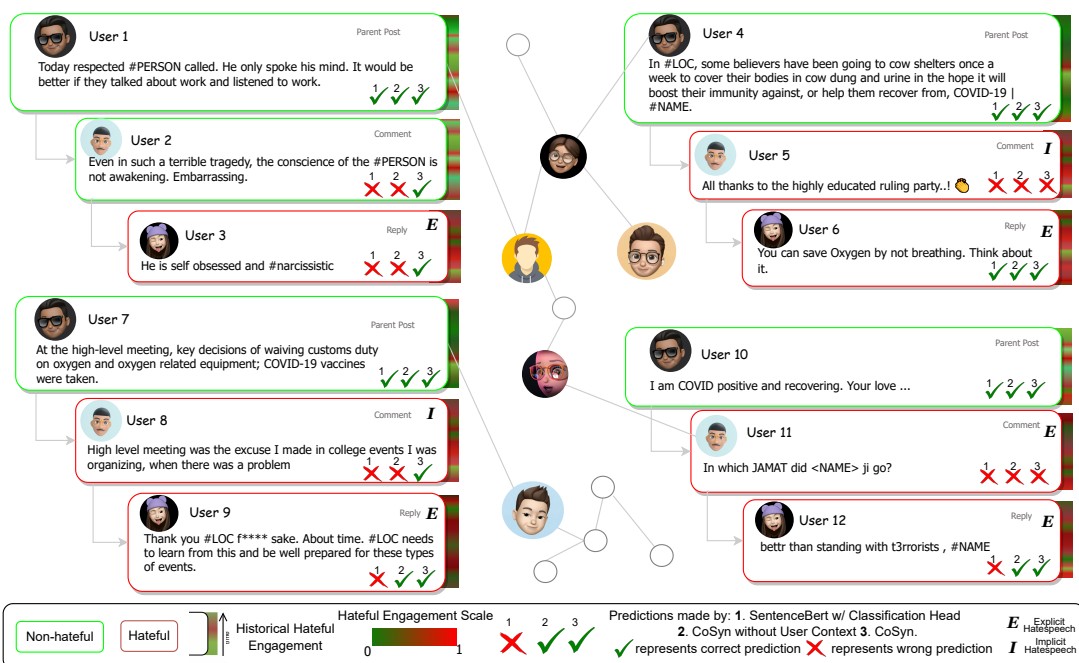

Figure 4: We study 4 conversation trees in the ICHCL dataset, including the prediction of different classifiers on the utterance to be assessed, the historical engagement of the author of the utterance, and the social relations between the different authors.

world knowledge through knowledge graphs (KGs) requires language-specific KGs, and short utterances in conversation trees make the retrieval weak. In the past, to classify context-sensitive hate speech in existing open-source datasets, researchers incorporated conversational context for hate speech classification (Gao and Huang, 2017; Pérez et al., 2022). However, these systems employ naive fusion and fail to leverage structural dependencies between utterances in the conversation tree. Another line of work explores author profiling using community-based social context (connections a user has with other users) (Pavlopoulos et al., 2017; Mishra et al., 2018). However, the representations are not learned end-to-end and employ naive fusion.

Hyperbolic networks have been explored earlier for tasks that include modeling user interactions in social media, like suicide ideation detection (Sawhney et al., 2021b, 2022b), fake news detection (Grover et al., 2022), online time stream modeling (Sawhney et al., 2021a), etc. All these works show that modeling the hierarchical and scale-free nature of social networks and data generated online benefits from modeling in the hyperbolic space over Euclidean space.

## 5 Conclusion

In this paper, we present CoSyn, a novel learning framework to detect implicit hate speech in online conversations. CoSyn jointly models the conversational context and the author's historical and social context in the hyperbolic space to classify whether a target utterance is hateful. Leveraging these contexts allows CoSyn to effectively detect implicit hate speech ubiquitous in online social media.

## Acknowledgement

This work was supported by ARO grants W911NF2310352 and W911NF2110026.

## Limitations

In this section, we list down some potential limitations of CoSyn:

1. Lacking world knowledge is one of CoSyn's potential limitations. The inclusion of world knowledge could serve as a crucial context, enhancing CoSyn's performance in this task (Sheth et al., 2022). We would like to explore this as part of future work.

2. Table 2 clearly demonstrates that CoSyn's effectiveness relies on the cohesive integration

of all its components. Therefore, as a direction for future research, our focus will be on enhancing the performance of individual components.

## Ethics Statement

Our institution's Institutional Review Board (IRB) has granted approval for this study. In the annotation process, we took precautions by including a cautionary note in the instructions, alerting annotators to potentially offensive or distressing content. Annotators were also allowed to discontinue the labeling process if they felt overwhelmed.

Additionally, in light of the rapid proliferation of offensive language on the internet, numerous A0-based frameworks have emerged for hate speech detection. Nevertheless, a significant drawback of many current hate speech detection models is their narrow focus on explicit or overt hate speech, thereby overlooking the identification of implicit expressions of hate that hold equal potential for harm. CoSyn could ideally identifying implicit hate speech with remarkable accuracy, preventing targeted communities from experiencing increased harm online.

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

## A Dataset Details

**Reddit.** The Reddit hate speech intervention dataset (Qian et al., 2019) has 5,020 conversations, including 22,324 comments. On average, each conversation consists of 4.45 comments, and the length of each comment is 58.0 tokens. 5,257 comments are labeled hate speech, and 17,067 are labeled non-hate speech. Most conversations, 3,847 (76.6%), contain hate speech. Each conversation with hate speech has 2.66 responses on average, for a total of 10,243 intervention responses. The average length of the intervention responses is 17.96 tokens. User history, the timestamp for each post, and the username for each post were fetched using the Reddit API[2]. Dataset statistics can be found in Table 3. The Cohen's Kappa Scores for inter-annotator agreement for 3 pairs of annotators annotating for implicit hate speech were 0.78, 0.74, and 0.71.

**GAB.** The GAB (Qian et al., 2019) 11,825 conversations, consisting of 33,776 posts. On average,

---

[2]https://www.reddit.com/dev/api/

each conversation consists of 2.86 posts, and the average length of each post is 35.6 tokens. 14,614 posts are labeled hate speech, and 19,162 are labeled non-hate speech. Nearly all the conversations, 11,169 (94.5%), contain hate speech. 31,487 intervention responses were originally collected for conversations with hate speech, or 2.82 responses per conversation on average. The average length of the intervention responses is 17.27 tokens. User history was fetched using the GAB API[3]. The metadata for each fetched post provides information like the author and timestamp for a post. Dataset statistics can be found in Table 6. The Cohen's Kappa Scores for inter-annotator agreement for 3 pairs of annotators annotating for implicit hate speech were 0.85, 0.82, and 0.91.

**DIALOCONAN.** The DIALOCONAN dataset (Bonaldi et al., 2022) has more than 3K dialogical interactions between two interlocutors, one acting as the hater and the other as the NGO operator, for a total of more than 16K turns. The previous tweets of a particular user were considered to be the history for that. Dataset statistics can be found in Table 7. The Cohen's Kappa Scores for inter-annotator agreement for 3 pairs of annotators annotating for implicit hate speech were 0.79, 0.78, and 0.79.

**CAD.** The CAD dataset (Vidgen et al., 2021) is an annotated dataset of $\approx 25,000$ Reddit entries. The dataset is labeled with 6 conceptually different categories, including Identity-directed, Person-directed, Affiliation-directed, Counter Speech, Non-hateful Slurs, and Neutral. The dataset is also annotated with salient subcategories, such as whether personal abuse is directed at a person in the conversation thread or someone outside it. This taxonomy offers greater coverage and granularity of abuse than previous work. Each entry can be assigned to multiple primary and/or secondary categories. The dataset is also annotated in context, where each entry is annotated in the context of the conversational thread it is part of. Every annotation also has a label for whether contextual information was needed to make the annotation. User history, the timestamp for each post, and the username for each post were fetched using the Reddit API[4]. Dataset statistics can be found in Table 5. The Cohen's Kappa Scores for inter-annotator agreement for 3 pairs of annotators annotating for implicit hate speech were

---

[3]https://www.npmjs.com/package/gab-api
[4]https://www.reddit.com/dev/api/

---

**Algorithm 1** CoSyn: Implicit Hatespeech Detection

**Given:**
    N distinct conversation trees $T$ indexed by $T_n$
    L distinct users where each user is indexed by $u_l$
    historical utterances of user $u$, $H^u$
    each tree $T_n = \{t_0^{T_n}, \cdots, t_j^{T_n}, \cdots, t_J^{T_i}\}$    $\triangleright J^{T_n}$ utterances

**Initialize:**
    $e(.) \leftarrow e(J^{T_n})$    $\triangleright$Bias-invariant Encoder Training
    $\mathcal{U}_{u_l}^{hist} \leftarrow e(H^{u_l})$    $\triangleright$Encode historical utterances using HFAN.
    $\mathcal{U}_{u_l} \leftarrow HGCN(\mathcal{U}_{u_l}^{hist})$    $\triangleright$Modeling Personal User Context

**Process:**
    Considering $t_j^{T_n}$ belonging to user $u_l$
    $\mathcal{X}_j \leftarrow e(t_j^{T_n})$
    $h_j \leftarrow CSHT(\mathcal{X}_j, \mathcal{U}_{u_l})$    $\triangleright$CSHT
    $y_j^{gt} = \text{Softmax}(\text{MLP}([\log_o(h_j); \mathcal{X}_j]))$    $\triangleright$Final Prediction
**return** $y_j^{gt}$

---

0.65, 0.73, and 0.69.

**ICHCL.** The ICHCL dataset (Modha et al., 2021) (Identification of Conversational HateSpeech in Code-Mixed Languages) consists of hind-english code-switched Twitter conversations. The primary task is to identify comments and replies that can be considered acceptable when considered alone but may appear hateful, profane, or offensive when the context of a parent tweet is considered. Binary classification of such contextual posts was considered in this subtask. Around 7,000 code-mixed postings in English and Hindi were downloaded from Twitter and annotated in-house by the authors. User history was fetched using the Twitter API[5]. Dataset statistics can be found in Table 8. The Cohen's Kappa Scores for inter-annotator agreement for 3 pairs of annotators annotating for implicit hate speech were 0.72, 0.81, and 0.74.

**Latent Hatred.** The Latent Hatred dataset (ElSherief et al., 2021) is a hate speech dataset of 27K Gab messages annotated according to a 6-class taxonomy that includes White Grievance, Incitement to Violence, Inferiority Language, Irony, Stereotypes and Misinformation, and Threatening and Intimidation. Comments for every post, the user's user history, and the timestamp for each post were fetched using the Twitter API[6]. Dataset statistics can be found in Table 4. The Cohen's Kappa Scores for inter-annotator agreement for 3 pairs of annotators annotating for implicit hate speech were 0.91, 0.96, and 0.89.

## B   Baseline Descriptions

**Sentence-BERT w/ Classification Head** We use our bias-invariant Utterance Encoder trained and

---

[5]https://developer.twitter.com/en/docs/twitter-api
[6]https://developer.twitter.com/en/docs/twitter-api

| Split | Convs. | Comments | Replies | Hate | Non-Hate | Implicit | Explicit |
|---|---|---|---|---|---|---|---|
| Train | 13382 | 3604 | 6755 | 2065 | 11317 | 1160 | 905 |
| Val | 4461 | 1155 | 2314 | 653 | 3808 | 388 | 265 |
| Test | 4461 | 1181 | 2280 | 706 | 3755 | 400 | 306 |

Table 3: Dataset statistics for the Reddit dataset.

| Split | Convs. | Comments | Replies | Hate | Non-Hate | Implicit | Explicit |
|---|---|---|---|---|---|---|---|
| Train | 12485 | 418 | – | 3836 | 8649 | 3246 | 590 |
| Val | 4162 | 83 | – | 1309 | 2853 | 1096 | 213 |
| Test | 4162 | 76 | – | 2373 | 1789 | 2138 | 235 |

Table 4: Dataset statistics for the Latent Hatred Dataset.

| Split | Convs. | Comments | Replies | Hate | Non-Hate | Implicit | Explicit |
|---|---|---|---|---|---|---|---|
| Train | 13584 | 12704 | – | 2511 | 11073 | 1004 | 1507 |
| Val | 4526 | 4169 | – | 834 | 3692 | 375 | 459 |
| Test | 5307 | 4884 | – | 965 | 4342 | 405 | 560 |

Table 5: Dataset statistics for the CAD dataset

| Split | Convs. | Comments | Replies | Hate | Non-Hate | Implicit | Explicit |
|---|---|---|---|---|---|---|---|
| Train | 18300 | 5264 | 6948 | 6088 | 12212 | 2435 | 3653 |
| Val | 6100 | 1789 | 2336 | 1975 | 4125 | 737 | 1238 |
| Test | 6100 | 1809 | 2315 | 1976 | 4124 | 741 | 1235 |

Table 6: Dataset statistics for the Gab dataset

| Split | Convs. | Comments | Replies | Hate | Non-Hate | Implicit | Explicit |
|---|---|---|---|---|---|---|---|
| Train | 11675 | 2141 | 7393 | 5837 | 5838 | 1575 | 4262 |
| Val | 2436 | 458 | 1520 | 1218 | 1218 | 365 | 853 |
| Test | 2514 | 460 | 1594 | 1257 | 1257 | 336 | 921 |

Table 7: Dataset statistics for the DIALOCONAN dataset.

| Split | Convs. | Comments | Replies | Hate | Non-Hate | Implicit | Explicit |
|---|---|---|---|---|---|---|---|
| Train | 4643 | 3094 | 1483 | 2389 | 2254 | 1006 | 1383 |
| Val | 1348 | 684 | 397 | 695 | 653 | 388 | 307 |
| Test | 1097 | 849 | 483 | 452 | 645 | 184 | 268 |

Table 8: Dataset statistics for the ICHCL dataset.

inferred on utterances in isolation for this baseline.

**ConceptNet** Similar to Sentence-BERT but fused with world knowledge from ConceptNet inspired from (ElSherief et al., 2021; Lin, 2022).

**HASOC** (Farooqi et al., 2021) We use the system proposed by the winning solution in the HASOC 2021 challenge. The authors propose to train a transformer encoder by concatenating parent-comment-reply chains separated by the SEP token.

**Conc-Perspective** (Pavlopoulos et al., 2020) Similar to HASOC, but context concatenation is done only during inference and not during training.

**CSI** (Ruchansky et al., 2017) Capture, Score, and Integrate (CSI), originally proposed for fake news classification, implements a neural network to jointly learn the temporal pattern of user activity on a given utterance, and the user characteristic based on the behavior of users.

**GCAN** (Lu and Li, 2020) Graph-aware Co-Attention Network (GCAN), originally proposed for fake news classification, implements a neural network that uses the target utterance and its

propagation-based user's features.

**HYPHEN** (Grover et al., 2022) HYPHEN or Discourse-Aware Hyperbolic Fourier Co-Attention, proposed for social-text classification and incorporates conversational discourse knowledge with Abstract Meaning Representation graphs and employs co-attention in the hyperbolic space.

**FinerFact** (Jin et al., 2022) FinerFact, originally proposed for fake news detection, incorporates a fine-grained reasoning framework by introducing a mutual reinforcement-based method for incorporating human knowledge and designs a prior-aware bi-channel kernel graph network to model subtle differences between pieces of evidence.

**GraphNLI** (Agarwal et al., 2022) Graph-based Natural Language Inference Model (GraphNLI) proposes a graph-based deep learning architecture that effectively captures both the local and the global context in online conversation trees through graph-based walks.

**DUCK** (Tian et al., 2022) Rumour detection with user and comment networks (DUCK), similar to CoSyn, employs graph attention networks to jointly model the contents and the structure of social media conversation trees, as well as the network of users who engage in these conversations.

## C  Additional Details

**Model Parameters:** Sentence-BERT has $\approx$ 110M parameters with 12-layers of encoder, 768-hidden-state, 2048 feed-forward hidden-state and 8-heads. CoSyn has $\approx$ 18M parameters.

**Compute Infrastructure:** All our experiments are conducted on a single NVIDIA A100 GPU. An entire CoSyn training pipeline takes $\approx$ 40 minutes.

**Implementation Software and Packages:** We implement all our models in PyTorch [7] and use the HuggingFace [8] implementations of SentenceBERT.

**Potential Risks:** CoSyn relies on training data that may contain biases inherent in the data sources.

---

[7] https://pytorch.org/
[8] https://huggingface.co/

The detection system may inadvertently amplify or reinforce existing societal biases if these biases are not adequately addressed. For example, if the training data is biased towards specific demographics or ideologies, the model might exhibit unfair treatment or misclassification of certain groups, leading to potential discrimination or harm. Understanding the contextual nuances of language is a complex task. CoSyn might also be prone to over-generalization, potentially resulting in false positives or misclassification of non-hateful speech. The risk is that legitimate expressions of opinion or controversial yet non-hateful statements may be mistakenly flagged as hate speech. This could inadvertently lead to censorship or suppression of freedom of speech, limiting open dialogue and critical discussions on important social issues.