# OpenReview forum: "CoSyn: Detecting Implicit Hate Speech in Online Conversations Using a Context Synergized Hyperbolic Network"
_EMNLP/2023/Conference — EMNLP 2023 Main_

### Official Review · Reviewer_cZN1 · 2023-08-05

**Soundness:** 4

**Excitement:**

3: Ambivalent: It has merits (e.g., it reports state-of-the-art results, the idea is nice), but there are key weaknesses (e.g., it describes incremental work), and it can significantly benefit from another round of revision. However, I won't object to accepting it if my co-reviewers champion it.

**Missing References:**

Towards Suicide Ideation Detection Through Online Conversational Context
 https://dl.acm.org/doi/abs/10.1145/3477495.3532068 also see references by https://proceedings.mlr.press/v151/sawhney22a.html

**Paper Topic And Main Contributions:**

The authors study the problem of hatespeech detection on socal media. They build a neural network to capture implicit hate speech that uses user and conversational contexts. The authors experiment on 6 different hate-speech datasets and show improvements.

**Questions For The Authors:**

How long are the conversations on average? And what is the average number of tokens per comment?
Is it possible to feed these to LLMs if the context lengths fit?

**Reasons To Accept:**

1.	¬The proposed architecture, CoSyn is technically sound. The authors use hyperbolic learning to model the scale-free properties of the network.
2.	The experiments on 6 different datasets are good and the authors show performance improvements compared to baseline methods.
3.	The qualitative analysis is interesting and gives nice interpretations

**Reasons To Reject:**

1.	The paper is difficult to follow. Some of the motivations are not clearly explained. For example, it is unclear as to why the Fourier trasnsform is required.
2.	The idea of using conversation trees and hyperbolic learning is not novel. For example [1], study similar problems for different tasks. It is unclear as to what the technical novelties are compared to these recent works apart from changing the task.
3.	Furthermore, these recent baselines should also be adopted in the performance comparison.
4.	The architecture involves quite a few complex components which may add additional computational overload. An analysis on the complexity will be interesting to see. The hyperbolic operations are often time consuming and have unstable training process

**Reproducibility:**

3: Could reproduce the results with some difficulty. The settings of parameters are underspecified or subjectively determined; the training/evaluation data are not widely available.

**Reviewer Confidence:**

5: Positive that my evaluation is correct. I read the paper very carefully and I am very familiar with related work.

---

> ### Author Rebuttal · Authors · 2023-08-29
>
> Dear Reviewer cZN1,
>
> We thank you for the detailed analysis of our paper and the insightful comments. In what follows, we try to address all our comments point-by-point.
>
> ### Reasons To Reject
> > 1. The paper is difficult to follow. Some of the motivations are not clearly explained. For example, it is unclear as to why the Fourier transform is required.
>
> **Ans:** We apologize if you found it difficult to follow. While other reviewers explicitly appreciated the paper's clarity, we would like to know more about which Sections are difficult to follow so we can improve on them.
>
> As mentioned in Section 2.4, ​​we apply 2D DFT to the embeddings (1D DFT along the temporal dimension and 1D DFT along the embedding dimension) to capture the most commonly occurring frequencies (ideologies, opinions, emotions, etc.) in the historical utterances made by the user. Appendix A describes the DFT formulation, which fundamentally helps analyze frequency patterns and capture periodicities in embedded textual data. More precisely, applying the 2D DFT on embeddings, the model can efficiently identify the most recurring frequency components, and in our model, these repetitive patterns could correspond to consistently used themes, emotions, or ideologies in a user's historical utterances.
>
> > 2. The idea of using conversation trees and hyperbolic learning is not novel. For example, [1] studies similar problems for different tasks. It is unclear as to what the technical novelties are compared to these recent works apart from changing the task.
>
> **Ans:** We thank you for the question. We have highlighted novel contributions in our work compared to prior art, including [1]. We list them down as follows:
> 1. We do not use the original TreeLSTM used in [1]. Instead, we propose CSHT, which introduces a novel context interaction mechanism that clearly captures the interplay between user and conversational contexts. Unlike TreeLSTM cells, which take a single embedding as input, CSHT cells take in 2 inputs (both user and conversational contexts) and independently assess the amounts of information to be retrieved from both.
> 2. We are the first to introduce the idea of leveraging multiple diverse contexts for implicit hate speech detection and discuss why such contexts are important for the task (last paragraph of Section 1 and Section 2). Subsequently, we propose the CoSyn architecture, which introduces a novel learning methodology leveraging all these contexts intuitively and clearly. CoSyn, as we see in Table 1, improves performance over most baselines.
> 3. None of the individual components in CoSyn have been directly proposed in prior art. We are inspired by prior art for some of our operations in the components, but the formulation of each individual component is novel.
>
> We do not propose hyperbolic space learning as our major novelty. Instead, we show how different modules in CoSyn come together to solve the implicit hate speech detection problem. Modeling in the hyperbolic space just helps us consider the scale-free properties of social media posts and gives CoSyn an extra boost.
>
> > 3. Furthermore, these recent baselines should also be adopted in the performance comparison.
>
> **Ans:** We thank you for the suggestion. As requested, we compare the performance of CoSyn with the two suggested methodologies from the literature [1,2] and show the results in the following Table 1. We will also include these in the final version of our paper.
>
> |      |              | Overall      | |        |    Implicit   |      |        |
> |--------------|---------------|-------|----------|--------|--------|-------|--------|
> |              | F_1           | P     | R        | F_1    | P      | R     | C. F_1 | R. F_1 |
> | Reddit    |        |       |          |        |        |       |        |        |
> | HCN          | 56.01         | 55.64 | 56.38   | 30.03  | 31.18  | 28.96 | 26.90   | 21.67  |
> | MRIL         | 58.91         | 57.30  | 60.61    | 31.76  | 30.12  | 33.58 | 36.67  | 32.05  |
> | **CoSyn (ours)** | 76.23        | 79.07 | 73.58   | 81.12 | 80.15  | 82.12 | 60.23 | 59.12  |
> |              |               |       |          |        |        |       |        |        |
> |  GAB            |           |       |          |        |        |       |        |        |
> | HCN          | 52.51         | 53.88 | 51.21    | 41.60   | 46.11  | 37.89 | 24.25  | 28.97  |
> | MRIL         | 61.21         | 60.75 | 61.67    | 40.92  | 33.65  | 52.19 | 42.01  | 33.89  |
> | **CoSyn (ours)** | 66.71        | 64.43 | 69.15    | 45.00     | 37.01  | 57.38 | 46.22  | 38.29 |
> |  CAD            |           |       |          |        |        |       |        |        |
> | HCN          | 34.63         | 37.18 | 32.41   | 27.65  | 22.30   | 36.37 | 23.87  | -      |
> | MRIL         | 34.41         | 35.27 | 33.59    | 40. 65 | 42.90   | 38.62 | 21.35  | -      |
> | **CoSyn (ours)** | 73.26         | 70.07 | 76.75    | 57.59  | 59.27  | 56.01 | 38.19  | –      |
> |              |               |       |          |        |        |       |        |        |
> |   ICHCL           |         |       |          |        |        |       |        |        |
> | HCN          | 65.36         | 60.45 | 71.14    | 27.66  | 27.03 | 28.32 | 25.03  | 27.99  |
> | MRIL         | 66.71         | 67.87 | 65.59    | 30.22  | 31.65  | 28.91 | 29.01  | 29.89  |
> | **CoSyn (ours)** | 89.53         | 90.55 | 88.53    | 46.03  | 46.26  | 45.82 | 46.85  | 45.89  |
> |    DIALOCONAN          |    |       |          |        |        |       |        |        |
> | HCN          | 39.65         | 34.67 | 46.30     | 25.31  | 23.89  | 26.91 | 22.19  | 15.90   |
> | MRIL         | 40.82         | 39.45 | 42.28    | 25.09  | 22.31  | 28.66 | 20.77  | 16.31  |
> | **CoSyn (ours)** | 51.02         | 52.92 | 49.25   | 52.98  | 54.67  | 51.39 | 48.77  | 54.00     |
> |              |               |       |          |        |        |       |        |        |
> |   Latent Hatred           |  |       |          |        |        |       |        |        |
> | HCN          | 50.47         | 52.39 | 48.68    | 30.11  | 32.96  | 27.71 | 21.32  | -      |
> | MRIL         | 57.32         | 54.88 | 59.98    | 40.16  | 37.80   | 42.83 | 37.61  | -      |
> | **CoSyn (ours)** | 64.65         | 61.92 | 67.63   | 53.28 | 47.66  | 55.49 | 40.12 | –      |
> |              |               |       |          |        |        |       |        |        |
>
>
> Table 2: Result comparison of HCN  and MRIL with CoSyn on 6 hate speech datasets.
>
>
> As we clearly see, CoSyn improves over HCN [1] by an average of 20.46% on overall performance and an average of 25.61% on implicit hate speech detection performance. On the other hand, CoSyn improves over MRIL [2] on overall performance by an average of 17% and an average of 21.2% on implicit hate speech detection.
>
> > 4. The architecture involves quite a few complex components, which may add additional computational overload. An analysis of the complexity will be interesting to see. The hyperbolic operations are often time-consuming and have an unstable training process
>
> **Ans:** We thank you for the insightful comments. We now analyze the complexity of CoSyn. We will add these details to the final version of the paper.
>
> **Complexity**:
>
> Bias Invariant Encoder: For a sequence of length t tokens, the complexity of encoding it is O(t^2), given that a transformer-based model is used.
>
> HFAN: For a given user, d past utterances with t tokens each are considered user history. The bias invariant encoder encodes these utterances, which follows with a complexity of O(d*t^2). A 2D Fourier transform is applied to the bias invariant representations, which costs O(d*t(log(t)+log(d)). The computational complexity of hyperbolic operations is asymptotically bounded by the complexity of calculating the vector norm, which is used to compute various projections and perform Mobius operations O(d*t). The complexity for the Mobius GRU, therefore, is then bound by the sequence length, which is O(t) in this case, once the hyperbolic projection has been accounted for. This is followed by Hyperbolic projection and computation of the Einstein mid-point, which is proportional to O(d^2), as followed by this discussion. O(d^2+t+d*t+d*t(log d + log t) + d*t^2).
>
> HGCN: The key computation in each layer of a GCN is the multiplication of the adjacency matrix and the node feature matrix. Consider a user graph with n nodes, and m dimensional features; the hyperbolic activations are of the order of O(n*m^2). The key computation in each layer of a HGCN is the multiplication of the adjacency matrix and the node feature matrix, which is O(n^2*m), but for sparse graph representations used by use, this can further be approximated to O(e*m) where e represents the number of edges in the graph. For L layers, therefore, the complexity of the HGCN is O(L*e*m + m*n^2).
>
> CHST: CHST is modeled with a tree-LSTM base. The complexity, therefore, depends on the number of nodes in the tree: O(p).
>
> The overall complexity combining the three components is thus: O(p) + O(e*m+n*m^2) + O(d^2+t+d*t+d*t(log d + log t) + d*t^2).
>
> Considering asymptotic complexity with respect to constant dimensions, the complexity can be expressed as a function of input size as O(p) + O(n). The stated complexities should be seen in light of the intricate mechanisms each architecture employs to fulfill its role effectively. We do not notice any major instability in CoSyn training. Due to the platform's limitations, we cannot provide an image that plots the loss against the training steps. However, we will add this to the final version of the paper.
>
> ### Questions For The Authors
> > 1. How long are the conversations on average? And what is the average number of tokens per comment?
>
> **Ans:** Thank You for the question. The conversation chains, on average (averaged across 6 datasets), are 17 utterances long with an average of 42 tokens per conversation. More details can be found in Tables 3,4,5,6,7 and 8 in the Appendix.
>
> > 2. Is it possible to feed these to LLMs if the context lengths fit?
>
> **Ans:** Feeding conversation chains to instruction-tuned LLMs might be an interesting direction of research we would like to explore as future work. However, no prior-work explores the prompts to use or the learning mechanism to be employed. Also, the maximum number of tokens per conversation across 6 datasets for our experiments is 13728, which might cross the context length of modern LLMs.
> Additionally, it should be noted that open-source LLMs might be unable to answer with their safety features. Additionally, there would be no clear way to feed user context (historical and social) within the conversational context.
>
> ### References
>
> [1] Sawhney, Ramit, et al. "Towards suicide ideation detection through online conversational context." Proceedings of the 45th international ACM SIGIR conference on research and development in information retrieval. 2022.
>
> [2] Sawhney, Ramit, et al. "Orthogonal Multi-Manifold Enriching of Directed Networks." International Conference on Artificial Intelligence and Statistics. PMLR, 2022.

---

### Official Review · Reviewer_4YGb · 2023-08-10

**Soundness:** 4

**Excitement:**

4: Strong: This paper deepens the understanding of some phenomenon or lowers the barriers to an existing research direction.

**Missing References:**

ICHCL 2021 dataset full paper:
S. Satapara, S. Modha, T. Mandl, H. Madhu, P. Majumder, Overview of the HASOC subtrack at FIRE 2021: Conversational hate speech detection in code-mixed language, in: P. Mehta, T. Mandl, P. Majumder, M. Mitra (Eds.), Working Notes of FIRE 2021 - Forum for Information Retrieval Evaluation, Gandhinagar, India, December 13-17, 2021, volume 3159 of CEUR Workshop Proceedings, CEUR-WS.org, 2021, pp. 20–31. URL: http: //ceur-ws.org/Vol-3159/T1-2.pdf

Benchmark ICHCL 2021 dataset

H. Madhu, S. Satapara, S. Modha, T. Mandl, P. Majumder, Detecting offensive speech in conversational code-mixed dialogue on social media: A contextual dataset and benchmark experiments, Expert Systems with Applications (2022) 119342. URL: https://doi.org/10. 1016/j.eswa.2022.119342


**Paper Topic And Main Contributions:**

This paper focuses on detecting implicit hate in social media conversational dialogue. The authors propose a model that exploits the user’s personal context, social context, and conversation context to predict hateful posts. The authors tested the proposed model Cosyn in six conversational hate speech datasets and claimed that their model outperformed the baseline of each dataset.

**Questions For The Authors:**

- Social media posts in Figure 1 might not be hateful conversations as per the definition by Schmidt and Wiegand (2019)). The utterance in the parent post might be cryptic.

- Authors choose the Hyperbolic Fourier Attention Network (HFAN) for author profiling or to learn historical context; for social context, they use the Hyperbolic Graph Convolution Network (HGCN). And Context Synergized Hyperbolic Tree-LSTM (CSHT) to learn the conversational context. The micro F1 score of the ICHCL 2021 dataset is 0.8953. I would like to advise the author to look at the paper by  (Madhu et al., 2023) where the reported macro F1 score is 0.892 using simple architecture (SentenceBERT and LSTM). I would like to ask the authors why they chose such a complex model for the experiment.

-Are the results reported by the authors statistically significant?

**Reasons To Accept:**

- The proposed model, CoSyn, beat the baseline of the six conversation hate speech datasets.

- Codes are available to reproduce the results


**Reasons To Reject:**

The authors did not mention the compelling reasons for designing the proposed model.

**Reproducibility:**

4: Could mostly reproduce the results, but there may be some variation because of sample variance or minor variations in their interpretation of the protocol or method.

**Reviewer Confidence:**

5: Positive that my evaluation is correct. I read the paper very carefully and I am very familiar with related work.

---

> ### Author Rebuttal · Authors · 2023-08-29
>
> Dear Reviewer 4YGb,
>
> We thank you for the detailed analysis of our paper and the insightful comments. In what follows, we try to address all our comments point-by-point.
>
> ### Reasons To Reject
> > 1. The authors did not mention the compelling reasons for designing the proposed model.
>
> **Ans:** Figure 2 and Section 3 describe in detail each component in CoSyn. We list the motivation behind each component below. We would also like to know if any section in the paper is not well-motivated; we will try our best to address and rewrite them in the final version of the paper.
>
> - User engagement on social media is often diverse and possesses scale-free properties. Thus, to account for the natural irregularities and effectively model a user’s personal historical context, we propose a novel Hyperbolic Fourier Attention Network (HFAN) that 1. Use Fourier Transform to model abstract frequencies or patterns appearing in the historical utterances by a user, and 2. Using hyperbolic representation to model the scale-free nature of historical discourse on social media.
> - HGCN has been used to account for the social context of a user defined by their social graph and the historical activity of the users. HGCN modifies the conventional GCN and performs neighbor aggregation using graph convolutions in the hyperbolic space to enrich a user's historical context representations learned through HFAN using social context.
> - To model the conversational context in conversation trees effectively, we propose Context-Synergized Hyperbolic Tree-LSTM (CHST). CSHT presents several modifications and improvements over Tree-LSTM, including (1) incorporating both the user's personal context and the conversation context while clearly capturing the interactions between them and (2) operating in the hyperbolic space, unlike the original TreeLSTM to account for the asymmetric properties of conversation trees on social media.
>
> CoSyn utilizes the advantages of established research while introducing innovative components that specifically consider user and conversation context. This is crucial for effectively identifying implicit hate speech.
>
> ### Questions For The Authors
> > 1. Social media posts in Figure 1 might not be hateful conversations as per the definition by Schmidt and Wiegand (2019)). The utterance in the parent post might be cryptic.
>
> **Ans:** We thank you for the insightful comment. We agree with your comments. Moreover, we don't think the parent post is hateful. Therefore, we mark it with a green border, with the key “Non Hateful.” Only the comment and the reply to the comment are hateful, which has a red border. We will make this more clear in the final version of our paper.
>
> > 2. Authors choose the Hyperbolic Fourier Attention Network (HFAN) for author profiling or to learn historical context; for social context, they use the Hyperbolic Graph Convolution Network (HGCN). And Context Synergized Hyperbolic Tree-LSTM (CSHT) to learn the conversational context. The micro F1 score of the ICHCL 2021 dataset is 0.8953. I would like to advise the author to look at the paper by  (Madhu et al., 2023) where the reported macro F1 score is 0.892 using simple architecture (SentenceBERT and LSTM). I would like to ask the authors why they chose such a complex model for the experiment.
>
> **Ans:** We thank you for the important question. We sincerely apologize for missing this baseline. Upon request, in Table 1, we present results on all 6 datasets for the system proposed by  (Modha et al., 2023).
>
> |      |          Overall     |       | |             |   Implicit    |       |        |
> |--------------|---------------|-------|----------|-------------|-------|-------|--------|
> |              | F_1           | P     | R        | F_1         | P     | R     | C. F_1 | R. F_1 |
> |  Reddit   |        |       |          |             |       |       |        |        |
> | Modha et Al  | 70.58    | 65.28 | 76.82    | 76.79 | 77.79 | 75.81 | 27.39  | 23.34  |
> | **CoSyn (ours)** | 76.23        | 79.07| 73.58    | 81.12       | 80.15 | 82.12 | 60.23  | 59.12 |
> |       GAB       |           |       |          |             |       |       |        |        |
> | Modha et Al  | 52.94   | 45.19 | 63.91    | 41.15 | 48.72 | 35.61 | 26.84  | 17.77  |
> | **CoSyn (ours)** | 66.71        | 64.43 | 69.15    | 45.00          | 37.01 | 57.38 | 46.22 | 38.29  |
> |      CAD        |           |       |          |             |       |       |        |        |
> | Modha et Al  | 50.47   | 55.72 | 46.13    | 51.77 | 50.67 | 52.91 | 36.63  | -      |
> | **CoSyn (ours)** | 73.26       | 70.07 | 76.75    | 57.59      | 59.27 | 56.01| 38.19 | –      |
> |     ICHCL         |         |       |          |             |       |       |        |        |
> | Modha et Al  | 82.01   | 84.63 | 79.55    | 39.56 | 39.00    | 40.14 | 39.81  | 39.72  |
> | **CoSyn (ours)** | 89.53         | 90.55 | 88.53   | 46.03      | 46.26 | 45.82 | 46.85  | 45.89 |
> |    DIALOCONAN          |    |       |          |             |       |       |        |        |
> | Modha et Al  | 46.45   | 40.06 | 55.26   | 36.96 | 29.44 | 49.64 | 47.62  | 33.82  |
> | **CoSyn (ours)** | 51.02  | 52.92 | 49.25   | 52.98       | 54.67 | 51.39 | 48.77  | 54.00     |
> |   Latent Hatred           |  |       |          |             |       |       |        |        |
> | Modha et Al  | 59.52   | 47.14 | 80.72    | 40.72 | 33.56 | 51.75 | 28.19  | -      |
> | **CoSyn (ours)** | 64.65         | 61.92 | 67.63    | 53.28       | 47.66 | 55.49 | 40.12  | –      |
>
>
>
> Table 1: Result comparison for  (Modha et al., 2023) and CoSyn on all 6 datasets.
>
>
> As we clearly see, (Modha et al., 2023) perform close to our SentenceBert baseline. Still, CoSyn outperforms it by an average of 9.9% on overall F1 performance and by an average of 8.18% on implicit hate speech F1. We hypothesize that this is because the architecture proposed by Modha et al. is similar to the SentenceBert baseline but with an extra LSTM head.
> These results also help prove that CoSyn, though more complex than most other architectures in terms of architecture design, effectively captures external context and helps improve implicit hate speech detection. The CoSyn architecture captures different aspects of the context by coordinating between the multiple individual components.
>
> > 3. Are the results reported by the authors statistically significant?
>
> **Ans:** We thank you for the question. As mentioned in the last line of Section 3.1, all scores in Table 1 are micro-F1 scores averaged across 3 runs with 3 different random seeds.

---

### Official Review · Reviewer_ArYr · 2023-08-13

**Soundness:** 4

**Excitement:**

4: Strong: This paper deepens the understanding of some phenomenon or lowers the barriers to an existing research direction.

**Paper Topic And Main Contributions:**

This paper provides a new method of implicit hate speech detection for conversational context by incorporating three aspects: (i) the author’s historical context, (ii) the author’s social context, and (iii) conversational context. Their method of inclusion of this information has been compared to numerous baselines and SOTA methods.
They extend the annotation scheme for six commonly used hate speech datasets by employing crowdsourcing annotation (with three raters per utterance).
Suppose this additional annotation for implicitness is made publicly available. In that case, it can be a valuable source for the researcher who tackles implicit hate speech detection. However, I have some concerns about the quality of the crowdsourcing annotations that I mentioned later on.


**Questions For The Authors:**

Will the additional implicit hate speech annotation be publicly available?

As the authors also claim, implicit hate speech is more challenging than explicit hate speech. On the other hand, several models, including the proposed one in Table-1, show better performance for implicit hate speech than the overall hate speech, which is counter-intuitive.  This could be originated from the annotation of the implicit hate speech conducted via crowdsourcing. I would like to hear from the authors what their opinion is.

Moreover, the details of the annotation study are missing. Is there any quality check for the annotators (e.g., via control questions)?


**Reasons To Accept:**

It is a very well-written and well-structured paper. The problem definition is formulated clearly. The methods and details of the architecture were explained transparently, contributing to its reproducibility.  Their evaluation seems sound and comprehensive: They evaluate their method on several datasets and compare their results with the performance of the existing methods on implicit hate speech.

**Reasons To Reject:**

I have stated my concerns in the “Questions for the Author section.” Despite its strength in the methodological part, this paper is missing an error analysis section, which I found very critical for NLP papers. Providing several examples where the model systematically fails (for both explicit and implicit cases), and a more profound outlook on the results in terms of the characteristics of the datasets would improve the impact of the paper.

**Reproducibility:**

4: Could mostly reproduce the results, but there may be some variation because of sample variance or minor variations in their interpretation of the protocol or method.

**Reviewer Confidence:**

3: Pretty sure, but there's a chance I missed something. Although I have a good feel for this area in general, I did not carefully check the paper's details, e.g., the math, experimental design, or novelty.

---

> ### Author Rebuttal · Authors · 2023-08-29
>
> Dear Reviewer ArYr,
>
> We thank you for the detailed analysis of our paper and the insightful comments. In what follows, we try to address all our comments point-by-point.
>
> ### Reasons To Reject
> > 1. Despite its strength in the methodological part, this paper is missing an error analysis section, which I found very critical for NLP papers. Providing several examples where the model systematically fails (for both explicit and implicit cases), and a more profound outlook on the results in terms of the characteristics of the datasets would improve the impact of the paper.
>
> **Ans:** We thank you for the feedback. Due to space constraints, we only provide one example with error analysis in Figure 4 on Page 8. Since the rebuttal does not allow additional materials, this is what we promise to do with the final version of the paper (with the extra page allotted to us):
> - Add one more example in the main paper similar to Figure 4.
> - Add at least 3 more examples in the Appendix similar to Figure 4.
>
> ### Questions For The Authors
> > 1. Will the additional implicit hate speech annotation be publicly available?
>
> **Ans:** Thank You for the question. Yes of course we will make them publicly available!
>
> > 2.  As the authors also claim, implicit hate speech is more challenging than explicit hate speech. On the other hand, several models, including the proposed one in Table-1, show better performance for implicit hate speech than the overall hate speech, which is counter-intuitive. This could be originated from the annotation of the implicit hate speech conducted via crowdsourcing. I would like to hear from the authors what their opinion is.
>
> **Ans:** We appreciate the reviewer's feedback and analysis. We notice that this phenomenon occurs only in approximately 7 instances across 6 datasets and 12 models (10% of the total experiments). The Implicit F1 score is a composite metric derived from three individual scores: Parent, Comment, and Reply F1 scores. Firstly, we notice certain cases where a data imbalance exists among these categories, which could affect the overall performance. Given that Parent tweets typically offer richer contextual cues, the model's superior performance in this category could have affected the final Implicit F1 score. Nevertheless, it's noteworthy that a significant performance gap persists between the overall and implicit Comment and Reply F1 scores, suggesting that detecting implicit hate speech remains a challenge across these tweet types.
> We concur with the reviewer's observation and attribute this phenomenon to other factors like data imbalance, model bias, or potential overfitting to specific characteristics.
>
> >  3. Moreover, the details of the annotation study are missing. Is there any quality check for the annotators (e.g., via control questions)?
>
> **Ans:** We sincerely apologize for missing these details. As mentioned in the paper in Section 3.1, we follow a two-stage annotation process where: During stage (1), we provide the annotators with the definition of hate speech and examples of explicit and implicit hate speech. During stage (2), we provide complete conversations and ask them to annotate implicit or explicit in a binary fashion. We will now elaborate on Step 1 (with the control questions) and include these details in the final version of our paper (with the extra page allotted to us).

---

### Meta-Review · Area_Chair_vzXE · 2023-09-19

**Recommendation:** 5

**Metareview:**

The paper under review presents a novel approach to implicit hate speech detection in conversational contexts. It incorporates the author's historical context, social context, and conversational context, comparing its methodology with various baselines and state-of-the-art (SOTA) methods. Additionally, the paper extends the annotation scheme for six widely used hate speech datasets through crowdsourced annotations. While there are some concerns regarding the quality of these annotations, the paper's clarity, thorough evaluation, and the superiority of the CoSyn model over baselines make it a valuable addition to the field, with potential significance for future research in implicit hate speech detection.

Three reviewers (ArYr, 4YGb, and cZN1) provided their feedback and insights on the paper. All reviewers acknowledge the paper's contributions to implicit hate speech detection and its comparative analysis against baselines and state-of-the-art methods. ArYr commends the paper's precise problem formulation, transparent explanations of methods, and comprehensive evaluation on multiple datasets. 4YGb highlights the success of the proposed model, CoSyn, in outperforming baselines across six conversational hate speech datasets and the availability of code for result reproducibility. cZN1 praises the paper's technical soundness, particularly using hyperbolic learning to model network properties and the interesting qualitative analysis providing interpretations.
Reviewer Concerns and Suggestions:

Some concerns, although addressable, have been raised by the reviewers as well. ArYr recommends adding an error analysis section and providing examples of where the model systematically fails for both explicit and implicit cases. 4YGb notes that the paper lacks a clear explanation of the compelling reasons for designing the proposed model. cZN1 asked for clarity regarding the use of Fourier transforms and the addition of a complexity analysis or a way to assess the computational load of the proposed model compared with baselines.

In summary, while the paper receives recognition for its contributions and strengths, the authors should address concerns related to error analysis and computational complexity to enhance the paper's impact. Future paper readers will also appreciate some changes in the manuscript to better motivate the inclusion of specific design elements like Fourier transform.

---

### Decision · Program_Chairs · 2023-10-07

**Decision:**

Accept-Main

**Comment:**

The paper under review presents a novel approach to implicit hate speech detection in conversational contexts. It incorporates the author's historical context, social context, and conversational context, comparing its methodology with various baselines and state-of-the-art (SOTA) methods. Additionally, the paper extends the annotation scheme for six widely used hate speech datasets through crowdsourced annotations. While there are some concerns regarding the quality of these annotations, the paper's clarity, thorough evaluation, and the superiority of the CoSyn model over baselines make it a valuable addition to the field, with potential significance for future research in implicit hate speech detection.

Three reviewers (ArYr, 4YGb, and cZN1) provided their feedback and insights on the paper. All reviewers acknowledge the paper's contributions to implicit hate speech detection and its comparative analysis against baselines and state-of-the-art methods. ArYr commends the paper's precise problem formulation, transparent explanations of methods, and comprehensive evaluation on multiple datasets. 4YGb highlights the success of the proposed model, CoSyn, in outperforming baselines across six conversational hate speech datasets and the availability of code for result reproducibility. cZN1 praises the paper's technical soundness, particularly using hyperbolic learning to model network properties and the interesting qualitative analysis providing interpretations.
Reviewer Concerns and Suggestions:

Some concerns, although addressable, have been raised by the reviewers as well. ArYr recommends adding an error analysis section and providing examples of where the model systematically fails for both explicit and implicit cases. 4YGb notes that the paper lacks a clear explanation of the compelling reasons for designing the proposed model. cZN1 asked for clarity regarding the use of Fourier transforms and the addition of a complexity analysis or a way to assess the computational load of the proposed model compared with baselines.

In summary, while the paper receives recognition for its contributions and strengths, the authors should address concerns related to error analysis and computational complexity to enhance the paper's impact. Future paper readers will also appreciate some changes in the manuscript to better motivate the inclusion of specific design elements like Fourier transform.